# The Autophagy Pathway: A Critical Route in the Disposal of Alpha 1-Antitrypsin Aggregates That Holds Many Mysteries

**DOI:** 10.3390/ijms22041875

**Published:** 2021-02-13

**Authors:** Celine Leon, Marion Bouchecareilh

**Affiliations:** INSERM, CNRS, U1053 BaRITOn, University Bordeaux, F-33000 Bordeaux, France; celine.leon@u-bordeaux.fr

**Keywords:** alpha-1 antitrypsin deficiency, Z aggregates, autophagy, ubiquitin–proteasome system (UPS), clearance, proteostasis

## Abstract

The maintenance of proteome homeostasis, or proteostasis, is crucial for preserving cellular functions and for cellular adaptation to environmental challenges and changes in physiological conditions. The capacity of cells to maintain proteostasis requires precise control and coordination of protein synthesis, folding, conformational maintenance, and clearance. Thus, protein degradation by the ubiquitin–proteasome system (UPS) or the autophagy–lysosomal system plays an essential role in cellular functions. However, failure of the UPS or the autophagic process can lead to the development of various diseases (aging-associated diseases, cancer), thus both these pathways have become attractive targets in the treatment of protein conformational diseases, such as alpha 1-antitrypsin deficiency (AATD). The Z alpha 1-antitrypsin (Z-AAT) misfolded variant of the serine protease alpha 1-antitrypsin (AAT) is caused by a structural change that predisposes it to protein aggregation and dramatic accumulation in the form of inclusion bodies within liver hepatocytes. This can lead to clinically significant liver disease requiring liver transplantation in childhood or adulthood. Treatment of mice with autophagy enhancers was found to reduce hepatic Z-AAT aggregate levels and protect them from AATD hepatotoxicity. To date, liver transplantation is the only curative therapeutic option for patients with AATD-mediated liver disease. Therefore, the development and discovery of new therapeutic approaches to delay or overcome disease progression is a top priority. Herein, we review AATD-mediated liver disease and the overall process of autophagy. We highlight the role of this system in the regulation of Z-variant degradation and its implication in AATD-medicated liver disease, including some open questions that remain challenges in the field and require further elucidation. Finally, we discuss how manipulation of autophagy could provide multiple routes of therapeutic benefit in AATD-mediated liver disease.

## 1. Introduction

Alpha 1-antitrypsin (AAT) is a serine protease inhibitor encoded by the *SERPINA1* gene. This protein is predominantly synthesized by liver hepatocytes and belongs to the serine protease inhibitor (PI) family, also known as Serpin [1]. The main function of this protein upon secretion into the circulation is to prevent lung tissue degradation by neutrophil proteases, such as neutrophil elastase, cathepsin G, and proteinase 3 [2].

Several mutations can affect the *SERPINA1* gene and lead to alpha 1-antitrypsin deficiency (AATD) [1]. This pathology is characterized by the accumulation of misfolded AAT proteins in the endoplasmic reticulum (ER) of hepatocytes [3], leading to a defective secretion of functional AAT [4]. This event results in a loss of the anti-protease activity of AAT and its ability to protect lung tissue from neutrophil enzyme-mediated degradation. This in turn predisposes patients with AATD to pulmonary symptoms, such as shortness of breath, wheezing, an increased risk of lung infections, and early-onset emphysema [2]. To follow up on the aforementioned AAT mutants, more than 100 variants have been identified and classified as follows: (i) null mutants: undetectable AAT serum levels due to nonsense mutations or frameshifts leading to a premature stop codon; (ii) deficient mutants: low AAT serum levels due to point mutations or small deletions [1].

Among all these mutants, the most commonly found deficiency allele in AATD is the deficient Z allele (p.Glu342Lys: allele frequency 0.0017) [5]. This mutant is also the most frequent variant (~95%) associated with liver disease [5]. The prevalence of this pathogenic variant accounts for 0.1% of the world’s population [6] and has a higher prevalence in Northern and Western Europe. This allele has been found in 4% of the Caucasian population in Northern Europe and in the United States, and it is estimated that 100,000 people carry this allele [7]. The Z mutant is caused by a single mutation at protein residue 342 leading to a lysine to glutamate substitution. This point mutation results in a misfolded protein that is retained within the ER of hepatocytes as two different forms: the soluble/monomer form [8,9], and also the insoluble/aggregate form [10]. Indeed, this single point mutation predisposes Z-AAT protein to polymerization and aggregation in the ER. This is histologically characterized by periodic acid-Schiff (PAS)-positive staining and the presence of diastase-resistant inclusion bodies (IB) in hepatocytes, the hallmarks of AATD-mediated liver disease on liver biopsies (Figure 1) [11]. Z-AAT aggregates play a distinct role in the disease pathology. From a molecular perspective, Z-aggregate accumulation is associated with the activation of several cellular stress pathways, including oxidative stress [12] and the nuclear factor-κB (NFκB) signaling pathway [13], which can lead to cell death. From a clinical and histological view, the accumulation of IBs increases as the fibrosis stage progresses in the liver of adult patients with AATD, and this accumulation can precede portal chronic inflammation and the development of liver fibrosis and cirrhosis [14]. Together these observations support the concept of a “toxic gain-of-function”, whereby Z-AAT retention within hepatocytes is responsible for liver disease [14].

Only 10% of patients with Z-AATD develop significant liver damage and 30%–40% develop intermediate liver fibrosis. This marked variability in the severity of liver disease associated with AATD could be in part explained by an impairment in the disposal pathways in patients with ZZ-AATD-mediated liver disease [15]. In agreement with this hypothesis, it has been shown that Z-AAT degradation is significantly slower in cells from patients with AATD and presenting with liver disease compared to those presenting without liver disease, suggesting an insufficient clearance of the Z variant in ZZ homozygous patients (defined as ZZ-AATD or ZZ) presenting with liver disease [16,17,18]. In contrary to the soluble forms of Z-AAT that are targeted for proteasomal degradation by the endoplasmic reticulum-associated degradation (ERAD) pathway [8,9], the insoluble/aggregate forms are disposed of via a different degradation pathway named autophagy [19,20]. Given that this pathway is the main machinery/process involved in Z-aggregate disposal, this route could be responsible for the liver damage associated with AATD. Therefore, in the next chapter we give an overview on the general process of autophagy before discussing the role of autophagy in AATD-mediated liver disease.

## 2. Autophagy

Autophagy is a cellular process through which cytoplasmic materials are delivered to the mammalian lysosome or to the yeast vacuole for degradation. The main functions of this pathway are the generation of degradation products and intracellular quality control by clearance of defective macromolecules or organelles [21]. Three types of autophagy have been described in mammalian cells:Micro-autophagyChaperone-mediated autophagy (CMA)Macro-autophagy

Micro-autophagy and CMA involve the direct uptake of cytosolic cargos, whereas macro-autophagy requires the formation of specific vesicles, known as autophagosomes, for the delivery of cargos to the lysosome.

### 2.1. Microautophagy

Micro-autophagy is the least characterized form. In yeast, this type of autophagy involves specific and nonspecific engulfment of cytoplasmic components (proteins or organelles) by direct invagination or protrusion of the vacuole membrane [22]. In mammals, micro-autophagy involves multivesicular bodies (MVB) that are formed at the surface of late endosomes. This is called endosomal micro-autophagy (eMI) [23]. The late endosome membrane contains the endosomal sorting complex required for transport (ESCRT) proteins, which are required for invagination, MVB formation, and the release of cytosolic proteins into the endosomal lumen [24]. This process can be both specific and non-specific, selectivity occurring through recognition of proteins containing KFERQ-like motifs by the chaperone protein called heat shock cognate protein 70-kDa (hsc70) [23].

### 2.2. Chaperone-Mediated Autophagy (CMA)

CMA is a selective form of autophagy by which specific cytosolic proteins are transported one-by-one across the lysosomal membrane for degradation [24]. Two major players are required in this process: hsc70 and lysosome-associated membrane protein type 2A (LAMP-2A). As mentioned above, hsc70 is a cytosolic chaperone protein that ensures selectivity through the recognition of KFERQ pentapeptide motifs of cytosolic target proteins. It targets these substrates to the lysosomal membrane and is likely to be involved in their unfolding, a process required for their translocation into the lysosome [25]. At the lysosomal membrane, target substrates bind to LAMP-2A, a lysosomal membrane protein required for the translocation of CMA substrates into the lysosomal lumen through their multimerization into a translocation complex. CMA can only degrade soluble proteins, and so cannot degrade organelles as does micro-autophagy.

### 2.3. Macro-Autophagy

Macro-autophagy, commonly called autophagy, is the most well-characterized form of autophagy and it can act either as a bulk process or a selective process. Selective autophagy is through autophagy receptors that recognize specific cargos, for instance: xenophagy for bacteria or viruses, aggrephagy for aggregated proteins, mitophagy for mitochondria. Macro-autophagy involves specific double-membraned vesicles called autophagosomes (Figure 2). The formation of autophagosomes is regulated by autophagy-related (ATG) proteins. Autophagosomes can sequester large amounts of cytoplasm, including parts of or entire organelles (Figure 2). To date, 42 ATG proteins have been identified and 16 of them belong to ″core″ autophagy proteins [21]. This core comprises five major complexes: (i) Unc-51 like autophagy activating kinase 1 (ULK1) complex; (ii) class III phosphoinositide 3-kinase complex 3 (PI3KC3); (iii) ATG9, the only transmembrane protein; (iv) WD repeat domain phosphoinositide-interacting (WIPI) proteins. A WD repeat protein is defined by the presence of four or more repeating units containing a conserved core of approximately 40 amino acids, usually ending with tryptophan-aspartic acid (WD); and finally (v) two ubiquitin-like conjugating systems: the ATG5-ATG12-ATG16L1 complex and the ATG8-PE complex [26]. These five complexes are involved in autophagosome formation, which can be separated into three steps: initiation, nucleation, and elongation-maturation [27] (Figure 2).

In addition to these core complexes, there are two other types of effector: the autophagy adaptors and the autophagy receptors. These are recruited by ATG8 family members (the γ-aminobutyric acid receptor-associated protein (GABARAP) subgroup and the light chain 3 (LC3) subgroup). Autophagy adaptors are likely to interact with GABARAP sub-family members through GABARAP interaction motif (GIM) on the convex autophagosomal membrane, regulating, for example, autophagosome formation (ULK1) or fusion with lysosomes (PLEKHM1) [28]. These autophagy adaptors are not degraded along with the cargos by the lysosome. It is probable that autophagy receptors interact with LC3 sub-family members through LC3 interaction motif/region (LIM/LIR) on the concave side (inner-autophagosomal membrane). These autophagy receptors recruit specific cargos and are degraded along with them after autophagosome-lysosome fusion. Autophagy receptors ensure that the autophagic process remains selective and are defined by their ability to link cargos to the autophagosomal membrane, leading to the engulfment of cargos by the autophagosome [29]. Alternatively, bulk autophagy serves to recycle building blocks to compensate for the lack of nutrients and is nonselective toward its substrates. Indeed, macro-autophagy was initially characterized as a bulk degradation pathway, but it is now clear that autophagy also contributes to intracellular homeostasis in non-starved cells by its selective degradation of cargo material, including aggregated proteins. Regarding AATD, this selective autophagy has a crucial role in the degradation of Z aggregates in the ER by the lysosome. This selective autophagy pathway is also named ER-phagy [30].

## 3. Autophagy and Alpha 1-Antitrypsin Deficiency

### 3.1. ER-Phagy

The ER is a dynamic structure and autophagy is important in its remodeling process [31]. As mentioned above, selective degradation of the ER (and thus its cargo) by autophagy is called ER-phagy (Figure 3). This process may not only contribute to the modulation of the shape of this organelle, but also to proteostasis control.

ER-phagy is mediated by different pathways (Figure 3), such as macro-ER-phagy: fragments of the ER are sequestered in autophagosomes that fuse with lysosomes; micro-ER-phagy (mainly observed during recovery from ER stress): the ER is targeted to the lysosome by engulfment; and finally, vesicular delivery: lysosomes can directly fuse with ER-derived vesicles [30]. However, to ensure selective ER-phagy all these processes require ER-phagy receptors. To date, six ER membrane proteins containing at least one LC3 (or GABARAP)-interacting region (LIR) have been identified as ER-phagy receptors in mammals: FAM134B [32], RTN3L [33], CCPG1 [34], SEC62 [35], TEX264 [36], and ATL3 [37]. All these receptors are expressed quasi ubiquitously, except for CCPG1 that is expressed predominantly in the pancreas, kidney, and liver. In addition to these ER-phagy receptors, it has recently been reported that p62 and DDRGK domain-containing protein 1-mediated ufmylation could also mediate ER-phagy [30,38].

ER-phagy and consequently its receptors are induced under different stimuli. Induction of ER-phagy following nutrient deprivation is mediated by FAM134B, ATL3, TEX264, and RTN3L [31,32,35,36]. These receptors induce the capture of ER sub-domains by autophagosomes. An additional two receptors, CCPG1 and SEC62, have been reported to mediate ER-phagy during ER stress and recovery from ER stress, respectively [34,35]. Finally, ER-phagy is also activated by proteasome-resistant misfolded proteins [39,40,41,42]. As previously described, misfolded soluble proteins in the lumen of the ER are targeted for degradation by the proteasome via the ERAD pathway [43]. However, some misfolded proteins, such as Z-AAT aggregates, fail to enter the ERAD pathway and accumulate in the ER. It has been reported that Z aggregates are able to activate this catabolic pathway and can be removed/degraded by distinct ER-phagy processes [44,45,46].

### 3.2. Disposal of Z Aggregates by Processes of ER-Phagy

The involvement of autophagy in AATD was first demonstrated by Teckman et al. [19]. The authors showed the accumulation of Z-AAT proteins in vesicles identified as autophagosomes according to several different hallmark criteria (ultrastructural and staining studies). Consolidating these results, Kaminoto et al. [44] provided genetic evidence of autophagy-mediated disposal of Z aggregates by showing a significant delay in disposal of insoluble Z-AAT in ATG5 ^−^/^−^ murine embryonic fibroblast cells. Since these experiments, several mouse and cell-based models, as well as in situ analyses from homozygous ZZ patient liver biopsies, have confirmed the presence of numerous autophagosomes, and more generally a role for autophagy in Z-AAT clearance [18,42,45,46]. Altogether, these results suggest that autophagic degradation plays a fundamental role in preventing the toxic accumulation of Z aggregates.

Z aggregates are able to induce autophagy through different routes. Indeed, conventional macro-autophagy inducers (including rapamycin, carbamazepine, or nor-urso-deoxycholic acid (norUDCA)) are able to induce autophagy through distinct cellular mediators (including PI3K/AKT, mTOR, or AMPK) and have been shown to reduce Z aggregate levels both in vitro and in vivo [45,47,48]. In addition, several autophagy factors that regulate lysosomal function and autophagy, such as the transcription factor EB (TFEB) master gene or the regulator of G signaling 16, have been found to be involved in Z-AAT disposal and may also represent mechanisms by which autophagy is activated in AATD [47,48].

However, Z aggregates can also undergo non-conventional autophagy. Recently, Fregno et al. described a novel mechanism of Z-AAT disposal by autophagy called ERLAD (ER-to-lysosome-associated degradation pathway) (Figure 3) [20]. This new pathway, that could be defined as a vesicular delivery pathway, involves calnexin and the engagement of the LC3 lipidation machinery by FAM134B, an ER-resident protein and ER-phagy receptor (Figure 3). In their model, Z-AAT aggregate delivery from the ER lumen to lysosomes for clearance does not require ER capture within autophagosomes. Rather, it relies on vesicular transport, where single-membrane, ER-derived, Z-AAT-containing vesicles release their luminal content within endolysosomes upon membrane-membrane fusion events. Lastly, Gelling et al. have identified sortilin 1 as a receptor for the delivery of aberrant Z proteins from the Golgi to lysosomes for degradation [49]. This pathway contributes to the intracellular disposal of this variant, suggesting that misfolded Z-AAT protein is subjected to appropriate quality-control throughout the entire secretory pathway [49].

### 3.3. AAT Inclusion Bodies and Autophagy

We previously mentioned that the hallmark for the detection of AATD in liver cells is the presence of PAS-positive and diastase-resistant globules, also named IBs. These structures represent dilated ER due to aggregated mutant protein retention [50,51]. However, these IBs are distinct compartments from the ER, even though they do stain positive for ER components (ribosomes and luminal ER protein, such as protein disulfide isomerase) and negative for lysosomal markers (such as LC3 and LAMP1). This is consistent with their identification as ER membranes and not lysosomes or autophagosomes [50,52]. In summary, IBs are neither classical autophagosomes nor lysosomes, but in fact a part of the ER. This compartment sheds to form IBs that are separate from the main ER compartment.

The segregation of Z-AAT to IBs could be considered as a protective cellular mechanism preventing ER stress [8]. Indeed, the failure to sequester Z-AAT in IBs and retention of Z-AAT in the ER causes cell shrinkage and induces a block in the secretory pathway at the step of protein exit from the ER [50]. Furthermore, Kaminoto et al. showed that the formation of IBs occurs in cells that elicit the classical autophagic response, but they also observed that the number of IBs increases in the autophagy-deficient ATG5 ^−/−^cell line [44]. Overall, these results suggest that IBs and autophagosomes belong to different pathways, and cells can sequester Z-AAT aggregates in IBs when all other pathways fail to regulate Z-AAT accumulation [53].

However, this protective mechanism presents some limits since the presence and the number of IBs have been associated with AATD-mediated liver damage. The presence of cells containing IBs in the livers of patients with AATD is considered to trigger a cascade leading to fibrosis and carcinogenesis [54], consistent with the theoretical model that disease occurs when proteotoxicity overwhelms mechanisms of proteostasis.

### 3.4. Open Questions Relative to the Role of Autophagy in AATD

Even if major progress has been made with respect to the role of autophagy in AATD, important questions still need addressing. For instance, could ATG-dependent effects be attributed to the vesicular delivery pathway rather than conventional autophagy? Moreover, it is currently unclear how the conventional autophagy pathway differs between macro-ER-phagy and the vesicular delivery pathway. In addition to FAM134B, CCPG1 could also be implicated in Z-AAT degradation. This ER-phagy receptor harbors long luminal tails which could play a role in the recognition of Z aggregates. Supplementary to the ER-phagy receptors already identified and analogous to those in mitophagy, cytosolic proteins could also detect changes in ER structure properties, such as curvature, modification, or ER membrane composition [30]. Why is the process of autophagy not capable of degrading IBs? How are the formed aggregates sequestered in IBs and why do they not undergo autophagy? This question is one of many regarding the role of autophagy in AATD that remains unclear. Further investigations are required to reveal the full significance of autophagy/ER-phagy in AATD.

## 4. The Proteasome Versus Autophagy in AATD: Two Closely Related Pathways

Based on a myriad of evidence, it is now obvious that crosstalk and interplay between the UPS and autophagy exist [55,56]. The complementary nature of both pathways has been well documented, particularly for proteinopathies observed in aging as well as in neurodegenerative disorders [55,56]. Thus, these two pathways share and eliminate common substrates [56], including the Z-AAT mutant [46], but also α-synuclein and amyloid-β, two aggregation-prone proteins involved in Parkinson’s disease and Alzheimer’s disease, respectively [57,58,59,60,61]. Additionally, these two machineries also share same factors/players; some enzymes of the ubiquitylation machinery, such as parkin, an E3 ligase that can be involved in both degradation pathways [62,63]. In summary, soluble substrates are degraded by the UPS, whereas much larger structures, such as large protein aggregates and insoluble complexes, are removed by autophagy. This is particularly true regarding the Z-AAT mutant. Both these machineries are involved in the clearance of this mutant; soluble Z-AAT being degraded by the UPS pathway, whereas autophagy is involved in the clearance of the insoluble and harmful forms of Z-AAT.

Nevertheless, how are these two pathways involved in, and communicate, in the induction of Z-AAT disposal? Based on yeast experiments, it appears that Z-AAT levels mediate the activation of these two pathways. The UPS pathway alone is activated when Z-AAT levels are low, and conversely autophagy is induced when Z-AAT levels are high and Z-AAT protein aggregation is initiated in the ER [64]. Polymerization/aggregation of Z-AAT proteins is likely to be a side-effect of ER retention of Z-AAT when the UPS system is overwhelmed or impaired [65], and subsequently autophagy is, rather, a secondary response [66]. 

In addition to these two major degradation pathways, it has been shown that the cell is able to dispose of Z-AAT aggregates by activating another degradation pathway that is independent of the proteasome-mediated mechanism and sensitive to tyrosine phosphatase inhibitors [67].

Nonetheless, even if hepatocytes do use both the UPS and autophagy “hand-in-hand” for Z-mutant degradation in order to protect themselves from Z-AAT proteotoxicity, in 10% of homozygous ZZ patients these degradation pathways are not sufficient [15]. In these cases, Z-AAT aggregates accumulate and trigger multiple signaling events, finally leading to cellular toxicity and death. Impairment of the ERAD pathway has been put forward for explaining the ″second hit″ involved in AATD-mediated liver damage [68]. No autophagy-related genes or factors have been identified as modifiers associated with AATD-mediated liver disease. Moreover, as we previously mentioned, accumulation of IBs increases as fibrosis stage progresses [14]. In this context, and in the light of this chapter, why is autophagy not activated in response to ERAD/proteasome impairment in order to prevent Z-AAT aggregate retention in IBs? Is the process of segregation of mutated AAT from the ER to IBs not recognized by and/or cannot activate autophagy? How does the cell decide how to segregate the Z variant and between which intracellular routes: ERAD/ERLAD/IB? Further studies on the mechanisms of IB formation and autophagy activation are likely to improve our knowledge on the mechanism by which cells sense and react to the accumulation of misfolded Z-AAT forms.

## 5. Targeting Autophagy for the Treatment of AATD-Mediated Liver Disease

We aforementioned that autophagy is activated when Z-AAT accumulates and it is specific for the degradation of insoluble forms of Z-AAT [44,69]. Accordingly, the autophagy pathway has been selected as a potential candidate for the treatment of AATD-mediated liver disease (Table 1). From the list of available autophagy-enhancing drugs, carbamazepine (CBZ) has been tested. CBZ is an FDA-approved anticonvulsant and mood stabilizer used in clinical practice. It has an extensive safety profile in humans. Hidvegi et al. showed that CBZ mediated a significant increase in the degradation of the insoluble forms of Z-AAT in vitro and in vivo through the increase of autophagic flux, even in cells already harboring activated autophagy [46]. In addition, the authors also demonstrated that CBZ mildly enhances proteasomal degradation of Z-AAT and has an independent action on non-proteasomal mechanisms for disposal of the soluble form of Z-AAT [46]. Thus, the effect of CBZ on Z-AAT clearance cannot be fully accounted for by the conventional disposal pathways (UPS and autophagy).

Even if CBZ increased the soluble form degradation rate, it had no effect on Z-AAT secretion given the serum concentrations of human Z-AAT were not affected by CBZ treatment in a murine model of Z-AATD. CBZ mediated a marked decrease in fibrosis in the livers of treated mice, but the doses used were 10-to-20 times higher than those used in humans. Nevertheless, given CBZ is extensively used in clinical practice, it has proceeded to a phase II/III trial for treatment at smaller doses of patients with AATD-mediated severe liver disease (NCT01379469). The results are currently not available.

The effects of rapamycin, another autophagy enhancer, were also tested on the same murine model of AATD [53]. A weekly dose of rapamycin was able to increase autophagic activity and consequently reduce the accumulation of Z-AAT aggregates in mouse livers. A decrease in markers of hepatocellular damage, including caspase-12 cleavage and fibrosis, were also demonstrated following rapamycin treatment. Interestingly, no difference in IB number or size was observed. This suggests that rapamycin treatment and autophagy act exclusively on Z-AAT aggregates, and not within IBs, and are involved in the degradation of aggregates before their incorporation into these structures. Even if these results are promising, there is as yet no clinical trial underway.

Other autophagy enhancers have also been identified by drug-screening platforms using the nematode *C. elegans* as a model [70] or hepatocyte-like cells derived from patient-specific induced pluripotent stem cells (iPSCs) in lines of patients with AAT deficiency [71] (Table 1). Based on the latter high-throughput drug screen, the authors were able to discover five hits, which consistently showed similar effects on the reduction of AAT accumulation in multiple patient’s derived hepatocyte-like cells. Interestingly, three drugs: lithium, carbamazepine and valproic acid among the final five hits were previously implicated as inositol-lowering autophagy-inducing agents and have been known to enhance the clearance of aggregate-prone proteins [72,73,74]. To go further, chemical modulation of autophagy upon metformin, trehalose or hydrogen sulfide treatments has been shown to have beneficial effects in some liver diseases such as in non-alcoholic fatty liver disease (NAFLD) [75]. Thus, it would be interesting to test their potential beneficial effects in AATD also (Table 1).

Additional promising treatments that do not directly target autophagy were also tested, such as norUDCA, which has anti-apoptotic effects among its many biological effects. Transgenic AATD mice treated with this compound showed a significant decrease in intrahepatic accumulation of Z-AAT. This was associated with an increase in autophagy and a decrease in hepatocyte cell death and liver damage [74,76].

Finally, other approaches to enhancing autophagy were examined, including gene therapy. Liver-directed gene transfer of TFEB, a master gene that regulates lysosomal function and autophagy, in transgenic AATD mice, resulted in a decrease in Z-AAT levels in the liver. This was associated with an increase in Z-AAT degradation mediated by enhanced autophagic flux (higher levels of LAMP1, enhanced SQSTM1/p62 degradation, increased LC3-I) [47]. The expression of TFEB also resulted in a reduction in IBs, apoptosis, and fibrosis in the livers. However, upon TFEB gene transfer, the authors detected a significant reduction of *SERPINA1* mRNA and Z-AAT monomer. This suggests that a different mechanism to that of increased degradation of Z-AAT in the autolysosomes must be involved.

## 6. Conclusions and Perspectives

In conclusion, AATD is a genetic disorder associated with an increased risk of liver disease in children and adults. Among the mutations responsible for AATD, the Z mutant is the most severe and common deficient variant. Homozygous ZZ patients may present with liver disease caused by the underlying retention and accumulation of Z aggregates in the ER of hepatocytes where Z-AAT is mainly synthesized and secreted. Some homozygous ZZ patients, but not all, may develop liver damage with variable severity, including cirrhosis or hepatocellular carcinoma. The cause of variability in susceptibility and severity of liver disease remains unclear, but there is growing evidence that this variability is due to modifiers. Given the role of autophagy in Z-AAT clearance by which the liver attempts to protect itself from proteotoxicity, this pathway is notably investigated. In the past years, major advances have been made regarding the identification of the role and regulation of autophagy in AATD-mediated liver disease. Nevertheless, several questions still remain unanswered and controversial. Further studies are required to elucidate these points and, in turn, potentially identify one or several modifiers that could hopefully open a new personalized approach to the treatment of AATD-medicated liver disease.

## Figures and Tables

**Figure 1 ijms-22-01875-f001:**
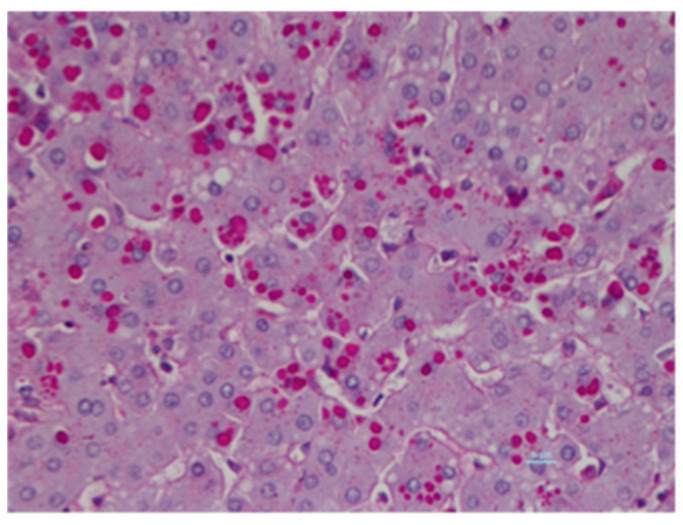
Z-alpha 1 antitrypsin (Z-AAT) aggregates in Inclusion Bodies. Human ZZ liver biopsy stained with hematoxylin-eosin and periodic acid-Schiff- (PAS)-Diastase digestion. The PAS-Diastase stains glycoproteins and consequently Z-AAT inclusion bodies in red.

**Figure 2 ijms-22-01875-f002:**
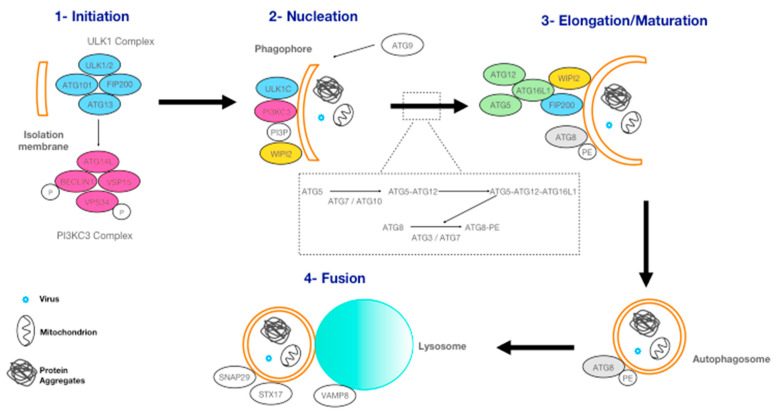
Autophagosome formation. The different steps are described precisely in Appendix A.

**Figure 3 ijms-22-01875-f003:**
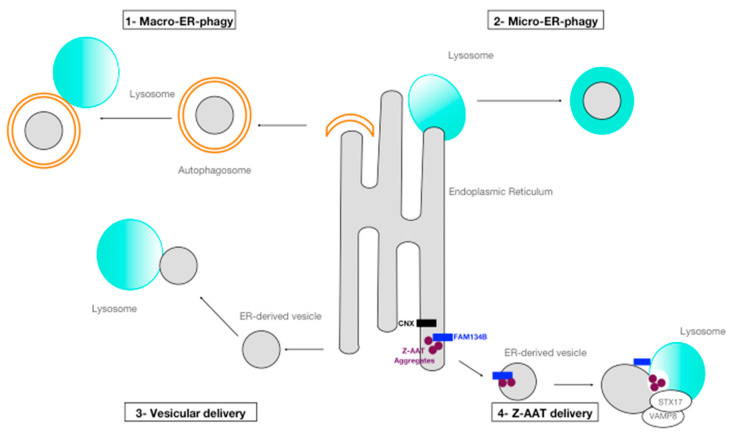
ER-phagy. (1) Macro-ER-phagy: fragments of the endoplasmic reticulum (ER) are enclosed in an autophagosome which fuses with a lysosome for degradation. (2) Micro-ER-phagy: a small portion of the ER is directly engulfed by inward invagination of the lysosomal membranes and then degraded. (3) Vesicular delivery: small vesicles containing misfolded proteins bud off from the ER and directly fuse with lysosomes. (4) Z-AAT delivery: calnexin (CNX) segregates proteasome-resistant Z-AAT aggregates in FAM134B-decorated ER subdomains. The SNARE proteins STX17 and VAMP8 regulate fusion of Z-AAT-containing ER-derived vesicles with lysosomes [20].

**Table 1 ijms-22-01875-t001:** Autophagy enhancer candidates for the treatment of alpha 1-antitrypsin deficiency- (AATD)-associated liver disease.

Treatment	Nature	Indication	Models	Results
Carbamazepine	Drug	Anticonvulsant/Mood stabilizer	Cell line and mouse model of AATD	Increase degradation of Z-AAT insoluble forms/Decrease of mouse liver fibrosis
			Drug screening on iPSCs from AATD patient	Reduction of AAT accumulation
Hydrogen sulfide	Drug	Autophagy inducer	High-fat diet-induced non-alcoholic fatty-liver disease (NAFLD) mouse model	Reduce steatosis and liver injury
Lithium	Drug	Antipsychotic	Drug screening on iPSCs from AATD patient	Reduction of AAT accumulation
Metformin	Drug	Anti-diabetic	Cell line and mouse model	Alleviates hepato-steatosis
Nor-orso-deoxycholic acid (norUDCA)	Drug	Anti-apoptotic effect	Mouse model of AATD	Increase autophagy /Decrease Z-AAT intrahepatic accumulation/Decrease of hepatocytes cell death and liver damages
Rapamycin	Drug	Immunosuppressor	Mouse model of AATD	Increase autophagy flux/Decrease Z-AAT aggregates/Decrease of fibrosis
Transcription factor EB (TFEB)	Gene	Regulation of lysosomal function and autophagy	Mouse model of AATD	Increase degradation of Z-AAT/Decrease the number of inclusion bodies (IBs)/Decrease fibrosis
Trehalose	Drug	Autophagy inducer	High-fat diet-induced NAFLD mouse model	Reduce steatosis and liver injury
Valproic acid	Drug	Anticonvulsant	Drug screening on iPSCs from AATD patient	Reduction of AAT accumulation

## Data Availability

Not applicable.

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
