# Peer review of "The Autophagy Pathway: A Critical Route in the Disposal of Alpha 1-Antitrypsin Aggregates That Holds Many Mysteries"

_ijms, 2021, doi:10.3390/ijms22041875_

Round 1
Reviewer 1 Report
This review about the role of autophagy in AATD is well written and clear. The first half of this MS is a very useful resume of autophagy, and the second half is more focussed on AATD, and the role of autophagy in this disease.
Despite these positive points, the MS can be improved by answering to several points:
- lines 50-51: the deficient Z allele is the most commonly found in AATD, but how many % of the mutants it represents ?
- lines 59-60: is it possible to add a picture showing the Z-AAT aggregates in the ER ? and/or the structure of AAT gene and this Z mutant ?
- line 74: please define ZZ-AATD and ZZ patient, I understand this is homozygous patient ?
- line 133: what is “WD” repeat domain ?
- Part 5 “targeting autophagy…”: in another review published in 2019 by Panda et al (Front Cell Dev Biol 7: 38), the authors wrote about the role of autophagy, and molecules able to favour autophagy in different kind of diseases, in part on liver diseases, CBZ and rapamycin were also described. Others, like valproate, lithium and trehalose are missing in this MS, and the authors should address their role in AATD in a new paragraph in this section.
- Abbreviation: please is it possible to get an alphabetical order ? some acronyms are missing like ZZ, PE…
Reviewer 2 Report
The work presented by Celine Leon and Marion Bouchecareilh, entilted “The Autophagy Pathway: a Critical Route in the Disposal of Al-2 pha 1-Antitrypsin Aggregates that Holds Many Mysteries” is a review that analyzes the main pathways for the elimination of Z-AAT aggregates with the aim of searching for new targets.
The paper is well-written, easy to follow, and contains two figures, one about macroautophagy and the other on endoplasmic reticulum-phagia. The paper deals in depth with the problems of the disease (pages 1 to 8) and goes more lightly into possible treatments (page 9).
In order to improve this last point, I suggest, for example, that the authors speculate on the reason why "Hidvegi et al. showed that CBZ mediated a significant increase in the degradation of both the insoluble and soluble forms of Z-AAT in vitro and in vivo through the increase of autophagic flux, even in cells already harboring activated autophagy" If CBZ is an autophagy-enhancing drugs and non an UPS activator.
I also suggest the inclusion of a summary figure of the main drugs in trial for the treatment of the disease, showing the specific targets, in order to improve point 5.
